# Central Roles of Glucosylceramide in Driving Cancer Pathogenesis

**DOI:** 10.3390/ijms26209879

**Published:** 2025-10-10

**Authors:** Xueheng Zhao, Manoj Kumar Pandey

**Affiliations:** 1Divisions of Pathology and Laboratory Medicine and Biomedical Informatics, Burnet Campus, 3333 Burnet Avenue, Cincinnati, OH 45229-3026, USA; 2Department of Pediatrics, University of Cincinnati College of Medicine, Cincinnati, OH 3333 Burnet Avenue, Cincinnati, OH 45229-3026, USA; 3Division of Human Genetics, Cincinnati Children’s Hospital Medical Center, MLC 7016S, Suite S4.206 (Office), Burnet Campus, 3333 Burnet Avenue, Cincinnati, OH 45229-3026, USA

**Keywords:** glucosylceramide, cancer metabolism, oncogenic signaling, cancer progression, anti-cancer therapy

## Abstract

Glucosylceramide (GlcCer), a central glycosphingolipid derived from ceramide, is increasingly recognized as a bioactive lipid that intersects with key metabolic, inflammatory, and oncogenic pathways. While its dysregulation has long been associated with lysosomal storage disorders such as Gaucher disease (GD), growing evidence implicates GlcCer in cancer initiation and progression, particularly within tumor-predisposing conditions. GlcCer modulates membrane microdomains, intracellular trafficking, and cell signaling, counteracting ceramide-induced apoptosis and promoting cellular survival. In cancer, aberrant upregulation of UDP-glucose ceramide glucosyltransferase (UGCG), the enzyme responsible for GlcCer synthesis, drives tumor growth, metastasis, and multidrug resistance through activation of pathways such as phosphoinositide 3-kinase/protein kinase B (PI3K/Akt), mitogen-activated protein kinase (MAPK), canonical Wnt pathway (Wnt/β-catenin), and nuclear factor kappa-light-chain-enhancer of activated B cells (NF-κB) pathways. Specific GlcCer species (e.g., C16:0, C18:0, C24:1) display tissue-dependent functions, adding structural specificity to their oncogenic potential. Moreover, emerging links between GlcCer metabolism and chronic inflammation, oxidative stress, and altered glucose utilization highlight its role as a metabolic node bridging inherited metabolic disorders and malignancy. This review integrates recent advances in GlcCer biology, emphasizing its roles in tumor-predisposing diseases and exploring its potential as a biomarker and therapeutic target in oncology.

## 1. Introduction

Glucosylceramide (GlcCer), also known as glucosylcerebroside, is the simplest glycosphingolipid and an essential bioactive lipid at the interface of membrane structure, metabolic regulation, and disease [1]. Composed of a ceramide backbone linked to a single glucose moiety via a β-glycosidic bond, GlcCer occupies a central role in sphingolipid metabolism and cellular homeostasis. Originally characterized in the 1920s in the spleens of Gaucher disease (GD) patients, GlcCer is now recognized as a key molecule embedded in various cellular membranes including the plasma membrane, Golgi apparatus, endosomes, and lysosomes, where it regulates membrane organization, trafficking, and cell signaling [2,3].

Synthesized in the Golgi apparatus via glucosylceramide synthase (GCS, also known as UDP-glucose ceramide glucosyltransferase—UGCG), GlcCer serves as the metabolic precursor for more than 90% of mammalian glycosphingolipids (GSLs), including complex gangliosides. These molecules are essential for mammalian development and viability, as evidenced by embryonic lethality in mice lacking UGCG [4,5]. Beyond its canonical role in glycosylation, GlcCer functions as a discrete signaling lipid, orchestrating critical processes including energy homeostasis, inflammatory responses, and membrane dynamics (Figure 1). 

Structurally, GlcCer is amphipathic, consisting of a hydrophilic glucose head group and a hydrophobic ceramide tail made of a sphingoid base and fatty acid [1]. Variations in the composition of these moieties such as acyl chain length, degree of hydroxylation, or type of sphingoid base (e.g., sphingosine, sphinganine, phytosphingosine) influence its biophysical properties and biological functions [6,7,8,9,10,11,12]. These structural differences impact GlcCer’s localization, turnover, and interactions within the lipid bilayer, including its enrichment in cholesterol-rich membrane microdomains known as lipid rafts [13].

In healthy cells, GlcCer homeostasis is tightly regulated by biosynthetic and catabolic pathways. It is synthesized from ceramide by UGCG and degraded predominantly in lysosomes by glucocerebrosidase (GBA1 as shown in Table 1), with contributions from non-lysosomal β-glucosidases (GBA2, GBA3) [14,15]. Disruption of this balance, as in GD, leads to pathological GlcCer accumulation, highlighting its physiological importance [16].

While much attention has been given to ceramide and its metabolites such as sphingosine and sphingosine-1-phosphate (S1P) in regulating cell death and proliferation, GlcCer has historically received less focus [34,35,36,37]. However, emerging evidence reveals that GlcCer metabolism is intimately linked to cancer progression [4]. Upregulation of UGCG is frequently observed in tumor cells, where it contributes to drug resistance, survival signaling, and malignant transformation. Transcription factors like specificity protein 1 (Sp1), modulated by oncogenic pathways, i.e., phosphoinositide 3-kinase/protein kinase B (PI3K/Akt) and mitogen-activated protein kinase (MAPK), drive UGCG expression, creating a metabolic shift that favors tumorigenesis [38,39]. Moreover, glucose availability, a hallmark of the tumor microenvironment, has been shown to directly influence GlcCer levels, tying cancer metabolism to lipid remodeling. Understanding the subcellular distribution and intracellular trafficking of GlcCer is also critical to cracking its role in cancer. After its synthesis in the Golgi, GlcCer is transported to other organelles and the plasma membrane through both vesicular routes and non-vesicular mechanisms mediated by lipid transfer proteins such as phosphatidylinositol 4-phosphate adaptor protein 2 (FAPP2) and glycolipid transfer protein (GLTP) [40,41,42]. Its trans bilayer movement is further regulated by flippases like ATPase phospholipid transporting 10D (ATP10D), whose dysfunction can exacerbate pathological lipid accumulation, as seen in GD and potentially in cancer cells [43].

This review examines the emerging roles of GlcCer metabolism in tumor-predisposing conditions, with a focus on its enzymatic regulation, subcellular localization, and involvement in oncogenic signaling. By integrating insights from developmental biology, metabolic disease, and cancer research, we aim to uncover the underappreciated role of GlcCer in tumorigenesis and assess its potential as a diagnostic marker and therapeutic target in cancer.

## 2. Glucosylceramide Architecture, Tissue Distribution, and Oncogenic Potential

Although GlcCer is a central component of glycosphingolipid metabolism, its biophysical properties and spatial distribution in mammalian systems remain less well characterized than those of other sphingolipids such as ceramide. This gap in knowledge is largely due to technical challenges in visualizing lipid species at subcellular resolution. Recent advances in chemical biology such as the use of photoactivatable (caged) lipids have enabled the functional dissection of GlcCer molecules within living cells [44,45]. These studies set the stage to reveal the role of individual GlcCer species in membrane behavior, dynamics, and downstream signaling functions.

The amphipathic nature of GlcCer allows it to associate strongly with cholesterol and participate in the formation of lipid rafts microdomains within membranes that serve as hubs for signaling, membrane trafficking, and cytoskeletal organization. Within the plasma membrane, GlcCer localizes preferentially to liquid-ordered phases, contributing to membrane rigidity and organization (Figure 2). Its dynamic distribution is also evident in intracellular compartments such as the Golgi apparatus, endosomes, and lysosomes, where it participates in vesicular trafficking and sorting.

The tissue-specific distribution of GlcCer is tightly regulated and reflects both local biosynthetic enzyme activity and functional demand. For instance, GlcCer is particularly enriched in macrophages, where it supports phagocytosis and endolysosomal trafficking [46,47,48]. GlcCer also affected the proliferation of Schwann cells and stimulated mitogenesis of murine epidermis [49,50]. It is also critical for keratinocyte differentiation and skin barrier formation, as UGCG expression is markedly upregulated during epidermal development [51,52]. In GD, the pathological accumulation of GlcCer within excess tissue and their resident immune cells, such as macrophages, dendritic cells, and T cells initiates a cascade of immunological disturbances. This buildup stimulates the release of proinflammatory cytokines and disrupts normal antigen presentation, triggering the chronic immune activation. Over time, this persistent inflammatory state contributes to the progressive damage of visceral organs, including the liver, spleen, lungs, and lymph nodes [15,53,54,55,56,57].

A parallel mechanism underlies the central nervous system (CNS) pathology of neuronopathic GD. In the CNS, deficiency of GCase and the accumulation of GlcCer—particularly in brain-resident cells such as microglia and neuron drive cellular activation and the excessive release of proinflammatory cytokines and chemokines. This inflammatory milieu contributes directly to neurodegeneration and cognitive impairment observed in neuronopathic GD [58,59,60,61,62,63,64].

Under physiological conditions, the GlcCer distribution in mammalian tissues is highly regulated. It is most abundant in the skin, spleen, intestine, and liver; moderately present in the lungs; and typically low in the brain, where its stereoisomer, galactosylceramide (GalCer), predominates as the principal glycosphingolipid [11,65,66]. 

Importantly, GlcCer exists not as a single molecule but as a family of structurally related species, differing in their fatty acyl chain length and degree of saturation. This structural heterogeneity is biologically meaningful [67,68]. In mammalian cells, typically GlcCer contains non-hydroxylated fatty acids including species with 16, 18, and 24-carbon fatty acids (C24:0 and C24:1) are predominant, accounting for more than half of the total GlcCer pool. However, tissue-specific variations exist: C16:0 and C22:0 GlcCer species are more prevalent in liver and adipose tissue, reflecting tissue-specific metabolic requirements and suggesting functional optimization for local physiological functions [68]. These patterns are shaped by the differential expression of ceramide synthase isoforms (CerS 1–6), each with distinct acyl-CoA substrate preferences. For example, CerS1 preferentially produces C18:0 ceramide, which has been implicated in apoptosis and tumor suppression, while CerS5 and CerS6 generate C16:0 ceramide, associated with cell survival and pro-tumorigenic signaling in certain cancers such as head and neck squamous cell carcinoma (HNSCC) [69].

These observations have important implications for cancer biology. The structural diversity of GlcCer species may contribute to divergent effects on tumor progression, depending on the tissue context and metabolic profile of the tumor. Elevated levels of specific GlcCer species particularly those derived from CerS5 and CerS6 may promote oncogenic signaling, modify plasma membrane lipid composition to enhance metastatic potential, and contribute to chemoresistance by altering drug uptake, efflux, or survival signaling pathways. Conversely, other species may suppress tumor growth or sensitize cells to apoptosis. The functional consequences of GlcCer composition, therefore, extend beyond membrane architecture to influence disease phenotypes, especially in tumor-predisposing conditions.

Moreover, GlcCer is not limited to endogenous biosynthesis. It can also be acquired from exogenous sources, including dietary intake and microbiota-derived lipids [70,71,72]. Plant-based GlcCer has been shown to cross the intestinal barrier, while microbial metabolism may influence local GlcCer levels in the gut, potentially impacting systemic lipid homeostasis and immune function [73,74]. These external sources may also interact with host metabolism in ways that modulate cancer risk, particularly in tissues exposed to the microbiota or dietary inputs.

Taken together, the distribution and composition of GlcCer in mammalian tissues are highly dynamic, context-specific, and biologically meaningful. Understanding how individual GlcCer species function within specific cell types and disease states, especially cancer may uncover novel mechanisms of lipid signaling and identify new therapeutic opportunities.

## 3. Metabolic Rewiring of Glucosylceramide Pathways in Cancer Development

Alterations in GlcCer metabolism have important implications for tumor predisposition, particularly in metabolic and lysosomal disorders. GD, the most common lysosomal storage disorder, results from mutations in the GBA1 gene, leading to deficiency of lysosomal β-glucocerebrosidase and pathological accumulation of GlcCer, especially within macrophages. This buildup promotes chronic inflammation, immune dysregulation, and oxidative stress, processes increasingly recognized as key contributors to elevated cancer risk. Elucidating the molecular pathways linking GlcCer accumulation to oncogenesis in these patients is essential for developing targeted preventive and therapeutic strategies.

Beyond lysosomal disorders, dysregulation of UGCG, the rate-limiting enzyme for GlcCer biosynthesis, is frequently observed across multiple malignancies and strongly correlates with tumor progression, metastatic potential, and multidrug resistance [75]. A detailed overview of GlcCer species implicated in specific cancer types is presented in Table 2.

Emerging evidence suggests that the distinct structural properties of GlcCer species, such as chain length and saturation, may underlie their functional specificity. Very long-chain fatty acids (VLCFAs, C ≥ 22) incorporated into GlcCer can alter membrane fluidity, influence lipid raft composition, and modulate signaling platform formation, whereas shorter or saturated species may preferentially interact with specific proteins or serve as precursors for downstream bioactive sphingolipids, thereby differentially regulating proliferation, survival, or metastatic pathways [89]. 

For example, in hepatocellular carcinoma (HCC), the mammalian target of rapamycin complex 2 (mTORC2) activation stimulates GlcCer production, driving steatosis and tumor development [86]. In cervical cancer, UGCG expression is upregulated in part by human papillomavirus (HPV) oncoproteins (E7 from HPV-16/18), which disrupt the Rb–E2F axis, promoting uncontrolled proliferation [84]. In chronic lymphocytic leukemia (CLL), C16:0 GlcCer specifically supports leukemic cell survival and proliferation, whereas C18:0 GlcCer has been implicated in GBA1-dependent liver cancer metastasis [77,80].

UGCG has emerged as a compelling therapeutic target in oncology, with robust support from preclinical models across multiple cancer types. Small-molecule UGCG inhibitors originally developed for GD and GBA-associated Parkinson’s disease provide important proof-of-concept for the pharmacological modulation of GlcCer metabolism (Table 3). Preclinical studies consistently demonstrate that inhibition of UGCG attenuates key malignant phenotypes, including suppression of tumor cell proliferation, inhibition of metastatic spread, reduction in immune cell infiltration, and impairment of tumor-associated angiogenesis (Table 4).

These findings suggest the need for advanced lipidomic approaches capable of resolving GlcCer species with high structural precision across diverse cancer contexts. Such approaches will be essential for elucidating how specific glycosphingolipid species contribute to oncogenic signaling, thereby enabling a mechanistic understanding of how structural diversity translates into functional specificity.

Collectively, the evidence supports a causal role for UGCG-driven GlcCer accumulation in oncogenesis and positions GlcCer metabolism as a promising, druggable node within metabolic signaling network of cancer [91,101,102,103,104,105]. Nevertheless, translating these insights into clinical application remains challenging. Tumor-selective drug delivery is essential to maximize therapeutic efficacy while minimizing systemic toxicity—a known limitation in glycosphingolipid-targeting approaches [106,107,108]. Moreover, cancer cells exhibit considerable metabolic flexibility; inhibiting GlcCer synthesis can trigger compensatory shifts to alternative sphingolipid pathways, reducing drug effectiveness [108,109]. This metabolic rewiring underlines the need for systems-level preclinical modeling. Finally, because glycosphingolipids are critical for maintaining epidermal integrity [110,111,112,113], neural development [114], and immune regulation [115,116,117,118], therapeutic strategies must carefully balance antitumor activity with preservation of normal physiological functions [5,119,120,121,122,123,124].

## 4. Glucosylceramide-Driven Crosstalk Between Energy Metabolism, Inflammatory Signaling, and Oncogenesis

GlcCer has emerged as more than a metabolic intermediate; it is an active participant in interconnected regulatory networks linking cellular bioenergetics, oxidative stress, and oncogenic signaling. Its biosynthetic precursors UDP-glucose, palmitoyl-CoA, and serine are intimately coupled to core energy metabolism, prominence a direct interface between lipid biosynthesis and mitochondrial function (Figure 2).

In cancer, this pathway is frequently hijacked to favor tumor survival. The conversion of pro-apoptotic ceramide into pro-survival GlcCer is enhanced, promoting chemoresistance, evasion of cell death, and sustained oncogenic signaling. Aberrant GlcCer trafficking and accumulation also modulate membrane microdomains and lipid-mediated communication with the tumor microenvironment, further supporting proliferation, inflammation, and metastasis. This “ceramide–GlcCer switch” represents a critical metabolic adaptation exploited by cancer cells and suggests UGCG and GlcCer metabolism as potential therapeutic targets.

Notably, although GCase primarily localizes to lysosomes, a portion of it is also found in mitochondria, where it supports respiratory activity and ATP generation [55]. In parallel, overexpression of UGCG enhances oxidative phosphorylation and glycolysis in breast cancer cells, whereas UGCG knockdown in cervical cancer cells impairs glucose uptake, lactate production, and ATP synthesis [76,84]. These metabolic defects are accompanied by disruptions in tricarboxylic acid (TCA) cycle intermediates, fatty acid oxidation, and phospholipid homeostasis, which are all the hallmarks of metabolic stress.

Beyond energy metabolism, GlcCer is tightly linked to oxidative stress pathways. ROS including superoxide anion, hydrogen peroxide, and hydroxyl radicals can initiate lipid peroxidation, alter membrane properties and influence UGCG activity in the Golgi apparatus. ROS also regulate UGCG expression under drug-resistant states, while defective autophagy exacerbates mitochondrial dysfunction, promotes ROS accumulation, and fosters tumorigenesis. Notably, ROS and GlcCer are linked in a feedback loop in cancer cells that promote tumor survival and drug resistance. While GlcCer can inhibit ROS in some contexts, related lipid signaling can also stimulate ROS-dependent immune functions. Furthermore, the fatty acid chain length of GlcCer influences its biological activity. In VLCFA-deficient cells, redox imbalance drives cell death, with ferroptosis emerging as a secondary consequence.

Signaling crosstalk further amplifies the oncogenic potential of GlcCer. Chronic mTORC1 activation documented in GD links GlcCer metabolism to inflammatory signaling and increased cancer susceptibility, including elevated melanoma risk in these GD patients [88,125,126,127,128,129]. Similarly, estrogen receptor signaling upregulates UGCG in hormone-responsive cancers, converging with MAPK cascades that promote proliferation, invasion, and metastasis [75,130]. 

At the transcriptional level, UGCG expression is tightly linked to PI3K/Akt and MAPK signaling, two core oncogenic pathways frequently activated in cancer. These cascades converge on the transcription factor Sp1, which drives UGCG transcription, thereby coupling oncogenic signaling with glycosphingolipid biosynthesis. This connection provides mechanistic evidence that UGCG upregulation is not a passive byproduct of malignant transformation but rather part of a coordinated program in which oncogenic signals reshape membrane composition, promote receptor clustering, and amplify proliferative and survival signaling [35,36,38,39]. Elevated UGCG expression has been associated with enhanced activity of downstream effectors, including nuclear factor kappa-light-chain-enhancer of activated B cells (NF-κB) and the canonical Wnt pathway (Wnt/β-catenin), which are known to promote tumor progression. Conversely, pharmacological inhibition of these signaling axes attenuates the tumor-promoting effects observed in UGCG-overexpressing models, indicating that altered glycosphingolipid metabolism can sustain oncogenic signaling in a feed-forward manner [75,97,131]. 

Strikingly, GlcCer has been shown to physically associate with β-1,4-galactosyltransferase 5 (B4GalT5), potentially enhancing glycosphingolipid biosynthesis and reinforcing tumor-promoting signaling [80,132]. Functionally, the conversion of ceramide to GlcCer attenuates ceramide-mediated growth arrest, apoptosis, and senescence, tipping the balance toward survival and proliferation [133]. Additionally, GlcCer-derived metabolites may act as secondary messengers in intercellular communication, embedding GlcCer deeper into tumor-supportive networks.

Collectively, these findings position GlcCer as both a metabolic sensor and a signaling hub in cancer biology. Its dual identity as a modifiable lipid species and a critical mediator of oncogenic signaling networks highlights its potential as a therapeutic target and biomarker, particularly in cancers linked to inherited or acquired metabolic dysregulation.

## 5. Discussion

Although the role of GlcCer in cancer biology is still emerging, growing evidence suggests that it serves as a key integrator of metabolic alterations and oncogenic signaling. By regulating glycosphingolipid synthesis and influencing membrane microdomain composition, GlcCer can modulate nutrient-sensing pathways, chemokines and growth factor receptor clustering, and downstream proliferative signaling. These effects mechanistically link altered cellular metabolism to oncogenesis [91,134,135,136]. At the molecular level, GlcCer synthesis depends on the availability of UDP-glucose, a direct product of cellular glucose metabolism. This dependency places GlcCer at the intersection of glycolytic flux, metabolic reprogramming, a hallmark of cancer and lipid signaling, positioning it as a key mediator through which glucose availability can influence oncogenic processes [137,138]. Glycosylation of ceramide to form GlcCer not only alters the biophysical properties of the membrane but also reshapes cellular signaling outcomes [139,140,141]. While ceramide acts as a tumor suppressor, promoting apoptosis and cell cycle arrest, its conversion into GlcCer attenuates these effects, supporting cell survival, proliferation, and therapy resistance [122,142,143].

Overall, the study of GlcCer has advanced significantly in recent years, yet its full implications in cancer biology particularly in the context of tumor-predisposing diseases remain underappreciated. Traditionally viewed as a precursor for complex glycosphingolipids, GlcCer is now increasingly recognized as a bioactive lipid, capable of influencing cell fate decisions, stress responses, and oncogenic transformation. This growing body of evidence calls for a re-evaluation of GlcCer not merely as a metabolic intermediate but as a critical player in tumorigenesis.

A central theme emerging from this review is that GlcCer metabolism is firmly linked to glucose availability, ceramide regulation, and oncogenic signaling pathways. Unlike its precursor ceramide, which promotes apoptosis and acts as a tumor suppressor, GlcCer serves as a metabolic switch that supports cell survival, immune evasion, and proliferation functions that are exploited in cancer. This metabolic shift is especially relevant in rapidly proliferating cancer cells, which display increased UGCG expression and elevated GlcCer levels, often correlated with multidrug resistance and poor prognosis [122,135,143,144,145,146,147,148,149,150,151].

GlcCer plays a central and multifaceted role not only within cancer cells but also in shaping the tumor microenvironment (TME) itself [80,102,152]. In cancer cells, dysregulated GlcCer metabolism reorganizes membrane microdomains, thereby influencing vesicular trafficking and the secretion of cytokines and chemokines that direct immune cell recruitment and activation—particularly macrophages, dendritic cells, and T cells—as observed in several malignancies [153,154,155,156]. At the same time, GlcCer metabolism within immune cells modulates tumor progression by driving the production of proinflammatory mediators and bioactive lipid metabolites. These changes establish self-reinforcing feedback loops that sustain chronic inflammation and strengthen a permissive, pro-tumorigenic niche [157,158,159,160,161,162].

The immune system, therefore, becomes an active architect of the TME, creating a dynamic ecosystem where immune cells are progressively reprogrammed to support rather than eliminate tumor growth [163]. Tumor cells exploit this plasticity by manipulating immune cell subsets including myeloid-derived suppressor cells (MDSCs), tumor-associated macrophages, dendritic cells, and regulatory T cells (Tregs) as well as soluble mediators such as vascular endothelial growth factor (VEGF), transforming growth factor-β (TGF-β), and cytokines like IL-10. Together, these factors polarize immune cells toward a pro-tumorigenic phenotype, forming a suppressive environment that fuels tumor growth, invasion, and metastasis [164,165,166,167,168,169,170,171].

Key drivers of MDSC expansion and recruitment into the TME include granulocyte colony-stimulating factor (G-CSF), macrophage colony-stimulating factor (M-CSF), and granulocyte-macrophage colony-stimulating factor (GM-CSF). Their functional maturation is promoted by cytokines such as IL4, IL6, IL13, and TNF, while mobilization into the tumor niche is orchestrated by chemoattractants including IL8, CCL2, and CXCL12 [171,172,173]. 

In GD, GlcCer accumulation due to lysosomal β-glucocerebrosidase deficiency results in the persistent expression of tumor-associated macrophage markers (e.g., CD163), CD206-positive bone marrow mesenchymal stromal cells, and activated CD4^+^T cells. This drives the release of a wide array of growth factors (G-CSF, M-CSF, GM-CSF, VEGF, TGF-β), chemokines (CCL2, CXCL12), and cytokines (IL4, IL6, IL13, TNF), along with IL9, TGF-β, and IL10, which collectively foster Tregs development [53,57,174,175,176]. 

This interplay between GlcCer accumulation and the emergence of a tumor-promoting microenvironment provides a compelling model for understanding how chronic metabolic imbalance and lipid dysregulation can prime tissues for oncogenesis. Consistent with this concept, a strong association between GlcCer storage and tumor occurrence has been reported in patients with GD [87]. Supporting this clinical observation, studies in a type 1 GD mouse model carrying the homozygous point mutation (Gba1 D409V/D409V) demonstrated that subcutaneous injection of murine melanoma cells (B16F10) led to markedly increased tumor growth and weight compared with control littermates [177]. 

These findings suggest that GlcCer accumulation not only reprograms immune responses but also actively conditions the tissue niche to favor tumor establishment. Beyond immune dysregulation, additional pathogenic processes including altered lysosomal function, metabolic reprogramming, and dysregulated sphingolipid signaling are now recognized as key contributors to the elevated incidence of hematologic malignancies and other cancers observed in GD [88,178,179,180,181,182,183,184]. 

Taken together, these glycolipid-driven mechanisms create a convergent network of signals that generates a permissive, pro-tumorigenic niche, supporting both tumor initiation and progression. 

The tissue-specific distribution and molecular diversity of GlcCer species introduce an additional layer of biological complexity to their role in cancer. Functional differences between species such as C16:0, C18:0, and C24:1 GlcCer indicate that not all GlcCer molecules exert uniform effects across cell types or pathological contexts (Table 2). These species-specific activities may arise from variations in membrane biophysical properties, selective enzyme interactions, or differences in protein–lipid binding affinities. For example, C16:0 GlcCer has been shown to promote leukemic cell proliferation in CLL models, supporting disease progression [77]. Similarly, C18:0 GlcCer has been implicated as a driver of GBA1-dependent liver cancer metastasis [80]. Notably, tumor growth can be sensitive to the regulation of VLCFAs; C ≥ 22, which serve as critical structural components of certain GlcCer species [89]. It has been found that VLCFA ceramides are preferentially converted to GlcCers by the action of UGCG. Inhibition of UGCG inhibits colon cancer cell growth in vitro and induced colon cancer development in mice [81]. GlcCers are important for colon barrier maintenance and activate the β-catenin pathway [82]. These data indicate that very-long-chain GlcCer are important for cellular signaling pathways that drive colon epithelial cancer development.

The role of UGCG as a central metabolic hub is particularly significant in cancer. Its transcription is directly regulated by key oncogenic signaling pathways, each contributing distinctly to GlcCer-mediated lipid remodeling [185]. Activation of the PI3K/Akt pathway enhances UGCG expression, promoting pro-survival lipid remodeling and chemoresistance [38]. The MAPK cascade stimulates UGCG transcription in response to mitogenic signals, supporting proliferation and metabolic adaptation [76,186]. Wnt/β-catenin signaling further upregulates UGCG, linking developmental and stemness programs to glycosphingolipid metabolism [75]. Finally, NF-κB activation sustains UGCG expression under inflammatory stress, integrating immune and stress signals with tumor-promoting lipid remodeling [75]. Collectively, these pathways converge to elevate UGCG levels, reinforcing GlcCer accumulation and establishing a metabolic environment favorable for tumor growth and survival [35,36,38,39,80]. Emerging evidence indicates that GlcCer metabolism can influence additional regulated cell death pathways, including autophagy and ferroptosis, which are increasingly recognized as critical in cancer progression and therapy resistance. Accumulation of GlcCer has been shown to modulate autophagic flux by altering lysosomal function and membrane composition, thereby affecting the balance between cell survival and programmed cell death. Similarly, GlcCer and related glycosphingolipids can impact ferroptosis susceptibility by regulating membrane lipid composition, redox homeostasis, and iron-dependent lipid peroxidation. These findings suggest that GlcCer not only serves as a metabolic and signaling hub for apoptosis but also intersects with other cell death pathways, creating additional avenues for therapeutic targeting in malignancies where conventional apoptosis-based therapies fail [187,188,189,190].

Importantly, pharmacological inhibition of these pathways has been shown to suppress UGCG-driven tumor growth in preclinical models (Table 3 and Table 4). 

Collectively, these findings emphasis the emerging potential of lipidomics platform, the stratified targeting of specific GlcCer subspecies, their biosynthetic enzymes, or their regulatory signaling pathways as a route toward more selective and effective anti-cancer therapies. In this context, UGCG stands out as both a biomarker of malignancy and a promising therapeutic target, particularly in cancers where ceramide/GlcCer metabolism has been rewired to tip the balance from apoptosis toward survival and proliferation.

## 6. Conclusions and Outlook

GlcCer has emerged as an active and versatile lipid mediator. It functions as a dynamic regulator at the interface of metabolism, signaling, and oncogenesis, with its structural diversity suggesting species-specific roles that have significant implications for cancer biology [80,148,191]. As advanced lipidomics analysis unmasks these molecular signatures, the field is poised to shift toward precision lipid targeting selectively disarming pathogenic GlcCer species while preserving essential lipid functions. The challenge ahead is to translate this biochemical complexity into actionable biomarkers and mechanism-based therapies, turning a once-overlooked lipid into a strategic target in cancer medicine. We hope this review serves as a foundation to spark greater interest within the research community in exploring this enigmatic glycosphingolipid in cancer development. Overall, decoding the structural and functional diversity of GlcCer may transform it from a metabolic bystander into a precision target for cancer prevention and therapy.

## Figures and Tables

**Figure 1 ijms-26-09879-f001:**
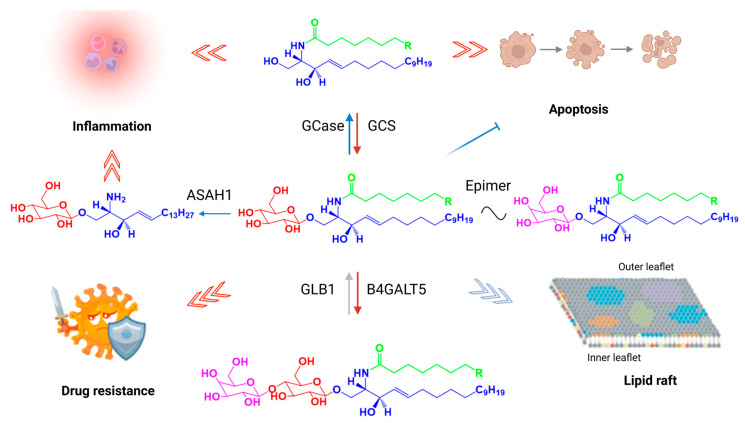
Structure, metabolism, and cancer-related rewiring of glucosylceramides (GlcCers). GlcCers are composed of a sphingosine backbone, a fatty acid chain of variable length and saturation, and a single glucose residue, generating molecular diversity that influences their biological functions. GlcCers are synthesized by the rate-limiting enzyme UDP-glucose ceramide glucosyltransferase (UGCG; locus 9q31, also known as glucosylceramide synthase), which transfers glucose from UDP-glucose to ceramide. Catabolism primarily occurs in lysosomes via lysosomal glucocerebrosidase (encoded by GBA, locus 1q21). GlcCers serve as precursors for complex glycosphingolipids, including lactosylceramide, globosides, and gangliosides—through sequential addition of sugars, sialic acids, sulfates, and other modifications.

**Figure 2 ijms-26-09879-f002:**
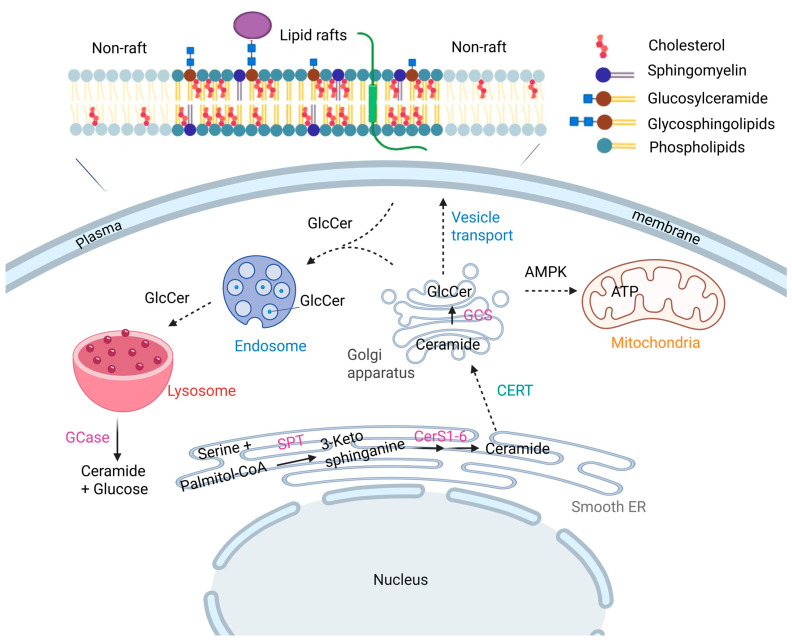
Glucosylceramide metabolic pathway and intracellular trafficking in cells, highlighting cancer-specific rewiring. De novo ceramide synthesis begins with the condensation of serine and palmitoyl-CoA, catalyzed by the serine palmitoyltransferase (SPT) complex, followed by acylation of dihydrosphingosine with variable-length fatty acids via six ceramide synthases (CERS 1–6). Ceramides are then transported to the Golgi, where a glucose residue is transferred to the primary hydroxyl group by UGCG, producing GlcCer. Newly synthesized GlcCer on the cytosolic face of the cis-Golgi is trafficked to the trans-Golgi, flipped to the lumenal side, and further glycosylated. GlcCers are then packaged into vesicles and transported to the plasma membrane, where they localize lipid rafts and participate in signaling, growth regulation, and apoptosis.

**Table 1 ijms-26-09879-t001:** Key enzymes involved in GlcCer metabolism and their functions.

Enzyme	UniProt ID (Human)	Mammalian Gene	Biochemical Function	References
Ceramide glucosyltranferase	Q16739 (EC 2.4.1.80)	UGCG (GCS)	Catalyzes the first glycosylation step in glycosphingolipid biosynthesis by transferring glucose from UDP-glucose to ceramide to form GlcCer.	[17,18]
Glucocerebrosidase (glucosylceramidase)	P04062 (EC 3.2.1.45)	GBA1	Hydrolyzes the β-glucosidic linkage of GlcCer to yield glucose and ceramide.	[19,20,21,22,23,24]
Acid ceramidase	Q13510 (EC 3.5.1.23)	ASAH1	Hydrolyzes ceramide into sphingosine and free fatty acids, thereby regulating sphingolipid homeostasis.	[24,25,26,27]
Galactosyltransferase orβ-1,4-galactosyltransferase 5	Q9UBV7 (EC 2.4.1.38)	B4GALT5	Transfers galactose from UDP-galactose to GlcCer to generate lactosylceramide (LacCer), a precursor for complex glycosphingolipids.	[28,29]
β-galactosidase	P16278 (EC 3.2.1.23)	GLB1	Lysosomal hydrolase that cleaves terminal β-galactose residues from glycoconjugates including LacCer and ganglioside GM1.	[24,30,31,32,33]

SAH1 (N-acylsphingosine amidohydrolase 1), B4GALT5 (beta-1,4-galactosyltransferase 5), GBA1 (glucosylceramidase beta 1), GCS (glucosylceramide synthase), GLB1 (galactosidase beta 1), GlcCer (glucosylceramide), LacCer (lactosylceramide), and UGCG (UDP-glucose ceramide glucosyltransferase).

**Table 2 ijms-26-09879-t002:** Glucosylceramide (GlcCer) species involved in tumor-predisposed diseases and cancers.

GlcCer Species	Sample Types	Signaling Pathway/Metabolic Dysregulation	Cancer Type	References
C16:0, C18:1, C18:0, total	Breast cancer tissues and cells	Glutamine metabolism, NF-κB and Wnt/β-catenin pathway	Breast cancer	[75,76]
C16:0, C24:1	Leukemic B-cell, human plasma	Pro-proliferation, mTOR, E2F1	Chronic lymphocytic leukemia	[77,78]
C24:1	Human plasma	Facilitate cancer cell survival	Oral squamous cell carcinoma (OSCC)	[79]
C18:0	Human liver cancer cells and tissue	Activation of the Wnt/β-catenin pathway	Liver cancer	[80]
Very-long-chain (C ≥ 22)	Mouse colon epithelial cell	Activation of the β-catenin pathway	Colon cancer	[81,82,83]
Total	Human cervical cancer tissue and cell line	Activation of the PI3K/AKT pathway	Cervical cancer	[84]
C12:0	Human liver tissue	Early-stage cancer biomarker	Intrahepatic Cholangiocarcinoma	[85]
Total	Mouse liver tissue	mTOR pathway	Hepatocellular carcinoma (HCC)	[86]
Total	Gaucher cells	Systemic inflammation with infiltration of immune cells	Melanoma in GD	[87,88]

E2F1 (E2 promoter binding factor 1), GD (Gaucher disease), GlcCer (glucosylceramide), mTOR (mammalian target of rapamycin), NF-κB (nuclear factor kappa-light-chain-enhancer of activated B cells), PI3K/AKT pathway (phosphoinositide 3-kinase/protein kinase B pathway), and UGCG (UDP-glucose ceramide glucosyltransferase).

**Table 3 ijms-26-09879-t003:** Glucosylceramide synthase (GCS, UDP-glucose ceramide glucosyltransferase, UGCG) inhibitor on preclinical mouse model.

GCS Inhibitor	Co-Treatment	Disease Treated	GlcCer Level	Disease Phenotype Change	References
GENZ-123346	None	Colon Cancer	Reduced	Significant lower tumor incidence in Genz-treated mice	[90]
threo-PPMP	Cisplatin	Head and neck cancer	Substrate ceramide increased	Sensitizes HNC preclinical tumor xenograft mouse model to cisplatin treatment	[91]
D-PDMP	None	Renal cancer	GlcCer increased and LacCer reduced	Marked reduction in tumor volume	[92]
Sinbaglustat	None	GM1 Gangliosidosis	Reduced in periphery	Decreased axonal damage and astrogliosis	[93]
Pyrazole Urea	None	Parkinson’s Disease	Reduced in CNS	Rescued lysosomal activity deficit and increased lysosomal hydrolysis activity	[94]
Eliglustat tartrate	None	GD	Reduced	Prevents GD-associated B-cell malignancy	[95]

CNS (central nervous system), D-PDMP (D-threo-1-phenyl-2-decanoylamino-3-morpholino-1-propanol), GD (Gaucher disease), GlcCer (glucosylceramide), HNC (head and neck cancer), threo-PPMP (D,L-threo-1-phenyl-2-palmitoylamino-3-morpholino-1-propanol), and UGCG (UDP-glucose ceramide glucosyltransferase).

**Table 4 ijms-26-09879-t004:** Glucosylceramide synthase (GCS, UDP-glucose ceramide glucosyltransferase, UGCG) inhibitors on human cell model of disease.

GCS Inhibitor	Co-Treatment	Disease Treated	GlcCer Level	Disease Phenotype Change	References
Miglustat	None	Colon cancer	Reduced	Marked arrest of the cell cycle in human colon carcinoma cells	[90]
threo-PPMP	Cisplatin	Head and neck cancer	N/A (substrate ceramide increased)	Increased cisplatin-induced cell death in HNC cells	[91]
GENZ-123346	Aripiprazole/cytostatic drugs	Hepatocellular carcinoma (HCC)	Reduced	Sensitizes HCC cells in therapy	[96]
Eliglustat	LAI	Melanoma	Reduced	Significantly inhibited tumor growth	[97]
Eliglustat	Anti-PD-1 antibody	Hematological malignancies and solid tumors	Reduced	Restore HLA antigen presentation	[98]
Eliglustat	None	Diffuse midline glioma (DMG)	N/A	Inhibited the proliferation of primary DMG cells	[99]
PDMP	SphK2 inhibitor	Lung cancer	N/A	Sensitize lung cancer cells to treatment	[100]

DMG (diffuse midline glioma), GlcCer (glucosylceramide), HCC (hepatocellular carcinoma), HNC (head and neck cancer), LAI (lysosomal autophagy inhibition), PDMP (1-phenyl-2-decanoylamino-3-morpholino-1-propanol), SphK2 (sphingosine kinase 2), and UGCG (UDP-glucose ceramide glucosyltransferase).

## Data Availability

No new data were created or analyzed in this study. Data sharing is not applicable to this article.

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
