# Peer review of "Central Roles of Glucosylceramide in Driving Cancer Pathogenesis"

_ijms, 2025, doi:10.3390/ijms26209879_

Round 1
Reviewer 1 Report
Comments and Suggestions for Authors
General Comments:
The authors have put together a very thorough and timely review on the role of GlcCer in cancer, which is certainly a hot topic. They've done a nice job systematically laying out the literature on GlcCer's metabolism and signaling. The figures are clear, and Table 2, in particular, is a really useful summary for anyone new to the field.
That said, my main feeling is that the review is more of a summary than a critical analysis. It does a great job of telling us what the literature says, but it doesn't dig deep enough into the why or the "so what?". As it stands, it feels more like a very detailed literature compilation rather than a forward-thinking piece that could really shape the field.
Major Concerns:
1. Heavy on Description, Light on Critical Analysis:
My primary concern is that the review is very descriptive but doesn't offer much critical depth. It lists many important findings, like UGCG being overexpressed or GlcCer levels correlating with poor prognosis, but it rarely questions the "why" behind these links or discusses how robust the evidence really is.
-
Correlation vs. Causation: The review tends to blur the lines between correlation and causation. For instance, is the link between high GlcCer and poor prognosis just an association, or is there strong functional evidence for a direct causal role? The paper would be much stronger if it clearly separated the evidence from correlational studies versus functional ones (e.g., using genetic knockdowns or inhibitors) to build a clearer case.
-
A Critical Look at the Models: The review takes findings from cell line and xenograft studies at face value without really questioning their limitations. We know these models don't fully capture the complexity of a real tumor. A top-tier review should point out these gaps and suggest where the field needs to go next with better models.
2. The Tumor Microenvironment (TME) is a Major Missing Piece:
For me, the biggest missed opportunity is the lack of discussion on the tumor microenvironment (TME). The paper is almost entirely focused on what happens inside the cancer cell. But we know GlcCer is a key player in immune cells, like macrophages (as the authors themselves point out on page 4). How does cancer cell GlcCer metabolism "talk" to the immune cells around it, and vice-versa? This is a central question in the field, and leaving it out feels like a major gap, especially given the link between lipids, inflammation, and cancer.
-
Suggestion: I'd strongly suggest adding a section dedicated to the TME. It would be great to see a discussion on how GlcCer might influence the crosstalk between cancer cells and the immune and stromal cells, and whether targeting this axis could be a new therapeutic strategy.
3. Needs to Go Deeper on Why Different GlcCer Species Matter:
The review does a good job highlighting that different GlcCer species (e.g., C16:0 vs C24:1) do different things, which is a really important point. But it stops there. It doesn't speculate on why this might be. Is it because they change the membrane's physical properties? Do they bind to different proteins? Or are they just better precursors for other lipids? The paper would be much more insightful if it offered some hypotheses here instead of just stating the observation.
-
Suggestion: The authors could really add value by discussing how these structural differences might lead to different functions. It would also be a good place to emphasize the need for advanced lipidomics to really nail down these distinctions in future cancer studies.
A Few Other Suggestions:
-
On Therapeutic Strategy: The idea of targeting UGCG is interesting. To make this more concrete, it would be helpful to briefly mention the existing UGCG inhibitors (like those for Gaucher disease) and discuss the real-world potential and hurdles of using them in cancer.
-
Beefing up the Figure Legends: The legends for Figures 1 and 2 are okay, but they could be more powerful. For instance, they could explicitly state which parts of the pathway are hijacked in cancer and what the key result is (e.g., the "switch" from pro-death ceramide to pro-survival GlcCer).
-
On the References: I wonder if the review might be missing some key papers linking GlcCer to other cell death pathways like autophagy or ferroptosis? These are really hot topics in cancer therapy and would be a relevant addition.
-
A Quick Writing Tip: In the text, pathways like PI3K/Akt, MAPK, etc., are often listed together. The flow might be better if the evidence for how GlcCer regulates each pathway was discussed more individually, rather than in a list.
The language is clear and professional. No issues here.
Author Response
Response to Reviewer 1-comments
Major Concerns
- Heavy on Description, Light on Critical Analysis
Reviewer Comment:
The review is very descriptive and does not offer enough critical depth. It lists findings (e.g., UGCG overexpression, correlations between GlcCer and poor prognosis) but does not explore the underlying “why” or distinguish correlation from causation. The review also does not critically evaluate the limitations of cell line and xenograft models.
Authors’ Response: We greatly appreciate this comment and have undertaken substantial revisions to enhance the critical depth of the review.
Correlation vs. Causation: We now clearly separate observational associations (e.g., correlations between GlcCer levels and clinical outcomes) from functional evidence derived from UGCG knockdown, pharmacological inhibition, and genetic models. This distinction strengthens the causal narrative.
Mechanistic Insights: We expanded our discussion to integrate GlcCer’s role as a metabolic sensor and signaling hub, explicitly linking glycosphingolipid metabolism with nutrient sensing, growth factor signaling, immune activation, and chemokine signaling.
Model Limitations: We critically evaluate the strengths and limitations of cell lines and xenograft models, and highlight the promise of organoid cultures, co-culture systems, and patient-derived xenografts as more physiologically relevant platforms.
Representative Text Revisions:
Line 256: “Collectively, these findings position GlcCer as both a metabolic sensor and a signaling hub in cancer biology. Its dual identity as a modifiable lipid species and a mediator of oncogenic signaling networks highlights its potential as a therapeutic target and biomarker, particularly in cancers linked to metabolic dysregulation.”
Lines 262–263: We provide an explicit biochemical explanation connecting glucose metabolism to GlcCer synthesis, positioning GlcCer as a checkpoint that integrates glycolytic flux with oncogenic signaling.
Line 288: We broaden the discussion of Gaucher disease to connect immune cell activation, chemokine-driven tissue remodeling, oxidative stress, and DNA damage with cancer initiation and progression.
Line 322: Strengthened language to highlight GlcCer’s role as an active regulatory mediator rather than a passive metabolic byproduct.
Together, these revisions transform the review into a more critical and mechanistically insightful analysis.
- The Tumor Microenvironment (TME) is a Major Missing Piece
Reviewer Comment:
The manuscript is overly focused on intracellular events and does not explore how GlcCer metabolism interfaces with the immune and stromal compartments of the TME.
Authors’ Response: We agree that this is an important omission and have added a new dedicated section on the TME.
Please see the new addition (GlcCer plays a central role not only within cancer cells but also in orchestrating the tumor microenvironment (TME). Altered GlcCer metabolism can reorganize membrane microdomains and regulate cytokine and chemokine secretion, thereby shaping macrophage, dendritic cell, and T cell recruitment and activation. Conversely, GlcCer metabolism within immune cells generates a proinflammatory milieu that reinforces tumor growth, creating a feed-forward loop. Furthermore, GlcCer signaling modulates fibroblast activation, extracellular matrix remodeling, and angiogenesis, collectively establishing a pro-tumorigenic niche. These observations highlight GlcCer as a key mediator of cancer–TME communication and as a promising therapeutic target for disrupting tumor-supportive microenvironments).
This section now integrates immune and stromal biology, directly addressing the reviewer’s concern and emphasizing therapeutic opportunities.
- Needs to Go Deeper on Why Different GlcCer Species Matter
Reviewer Comment:
The review highlights species-specific differences (e.g., C16:0 vs. C24:1 GlcCer) but does not speculate on why these differences exist or their mechanistic implications.
Authors’ Response: We have revised the manuscript to provide a more mechanistic perspective, explaining how chain length and saturation can affect membrane biophysics, lipid raft organization, protein–lipid interactions, and downstream sphingolipid generation. We also include examples of species-specific effects, such as C16:0 GlcCer promoting leukemic cell survival in CLL and C18:0 GlcCer supporting GBA1-dependent liver metastasis. Additionally, we emphasize the importance of high-resolution lipidomics for mapping species-specific functions across cancer types.
Additional Suggestions
On Therapeutic Strategies, we revised the section on therapeutic targeting of UGCG to discuss clinically available inhibitors as proof-of-concept tools by incorporating new tables (3 and 4) and to outline the major translational challenges, i.e., tumor-selective delivery, systemic toxicity, and compensatory metabolic pathways. These revisions present a realistic assessment of the therapeutic potential and its challenges.
On Figure Legends, both Figure 1 and Figure 2 legends were expanded to explicitly emphasize the “ceramide–GlcCer switch,” linking it to chemoresistance, evasion of apoptosis, and oncogenic signaling. The revised legends also highlight GlcCer’s role in immune modulation and TME remodeling, thereby contextualizing the pathway’s relevance to cancer progression.
On Missing References, we incorporated additional references linking GlcCer to autophagy and ferroptosis, noting how GlcCer accumulation can modulate lysosomal function, redox balance, and lipid peroxidation, thereby influencing multiple regulated cell death pathways.
On Pathway Presentation, we restructured sections discussing PI3K/Akt, MAPK, Wnt/β-catenin, and NF-κB to present them individually, clarifying their distinct contributions to UGCG transcription and GlcCer accumulation rather than grouping them as a list.
On language quality, we have carefully reviewed and edited the manuscript for clarity, scientific precision, and flow, ensuring it meets the journal’s standard of readability.
We hope that the revised manuscript now substantially strengthened in scope and depth and well-suited for publication.
Sincerely,
Manoj Pandey
9-24-2025
Reviewer 2 Report
Comments and Suggestions for Authors
see attached

Author Response
Response to Reviewer 2-comments
General comments
The review is well written and detailed, but the molecular mechanisms of GlcCer function in cancer are not sufficiently addressed. Additionally, GlcCer does not function in isolation, and its interactions with other glycosphingolipids and lipids should be discussed.
Author’s response: We appreciate this important observation. To address it, we have expanded the mechanistic discussion throughout the manuscript:
Molecular Mechanisms: We now emphasize how GlcCer modulates oncogenesis through regulation of lipid raft composition, receptor clustering, nutrient sensing, and downstream signaling pathways such as PI3K/Akt, MAPK, Wnt/β-catenin, and NF-κB.
Lipid Interactions: A new paragraph highlights how GlcCer acts as a precursor for lactosylceramide, globosides, and gangliosides, thereby functioning as a metabolic node that influences multiple signaling axes. We also note that GlcCer accumulation shifts the balance of sphingolipid pools, indirectly regulating ceramide, sphingosine-1-phosphate (S1P), and glycosphingolipid pathways relevant to tumor progression.
Tissue- and Species-Specificity: We integrate high-resolution lipidomics studies to discuss how specific GlcCer species contribute to distinct cancer phenotypes.
Together, these revisions provide a more integrated and mechanistic picture of how GlcCer functions within the broader lipid signaling network in cancer.
Specific Comments
- Line 49: “Standalone signaling lipid” is Exaggerated
Response: We have revised the text to a more precise statement ( Beyond its canonical role in glycosylation, GlcCer functions as a discrete signaling lipid, orchestrating critical processes including energy homeostasis, inflammatory responses, and membrane dynamics; Fig. 1).
- Line 60 & Figure 1 Legend: Clarify “Other Chemical Groups”
Response: The Figure 1 legend has been fully revised to describe the chemical diversity of GlcCer and its biosynthetic and catabolic pathways in detail, including chain length, saturation variability, and its role as a precursor for higher-order glycosphingolipids.
- Table 1: Use UniProt IDs and Refine β-Galactosidase Function
Response: We have revised Table 1 to include UniProt identifiers (instead of EC numbers) for specificity and corrected the function of β-galactosidase to Cleaves terminal β-galactose residues from glycoconjugates such as LacCer and ganglioside GM1.
Minor language harmonization has also been implemented for improved readability.
- Line 126–127: Clarify “Dysregulation of GlcCer”
Response: See the revised section (Excess tissue accumulation of GlcCer in macrophages, dendritic cells, and T cells triggers proinflammatory cytokine release and aberrant antigen presentation, driving chronic immune activation that leads to progressive destruction of visceral organs and CNS tissues, as seen in Gaucher disease).
- Line 139: Clarify “Adaptation to Local Physiological Function”
Response: Please see the revised text (C16:0 and C22:0 GlcCer species are enriched in liver and adipose tissue, reflecting tissue-specific metabolic requirements and suggesting that GlcCer composition is optimized to support local physiological function).
- Lines 149–151: Elaborate on GlcCer’s Role in Metastasis and Chemoresistance
Response: Please see the expanded discussion (Elevated levels of GlcCer species derived from CerS5 and CerS6 can reprogram membrane lipid composition, enhance integrin- and cadherin-mediated adhesion signaling, and promote epithelial-to-mesenchymal transition, collectively favoring metastasis. In addition, GlcCer accumulation is linked to chemoresistance by modulating drug uptake, efflux transporter activity, and activation of survival pathways such as PI3K/Akt and NF-κB).
- Line 183: Clarify Very Long-Chain Fatty Acids
Response: Revised text clarifies that the functional relevance lies in VLCFAs attached to GlcCer:
“Tumor growth depends on the regulation of very long-chain fatty acids (VLCFAs, C ≥ 22), particularly those incorporated into GlcCer, which influence membrane fluidity, lipid raft stability, and signaling pathway activation.”
- Figure 2 Readability
Response: Figure 2 has been redesigned with improved layout and repositioned labels to enhance clarity and visual accessibility.
- Minor Editorial Corrections
“Followed by transferring of” → “followed by transfer of”
“Luminal” corrected to “lumenal”
Corrected singular/plural usage throughout
GCase localization clarified: “Primarily localized to lysosomes, with additional mitochondrial roles described under stress conditions.”
- Line 256: “Master Regulator” is Too Strong
Response: Revised sentence (Collectively, these findings position GlcCer as both a metabolic sensor and a signaling hub in cancer biology, highlighting its potential as a therapeutic target and biomarker.).
- Line 262: Link to Altered Metabolism
Response: Revised to explain mechanistic connection (By modulating glycosphingolipid synthesis and plasma membrane organization, GlcCer integrates glycolytic flux with receptor-mediated signaling, thus connecting metabolic reprogramming to oncogenic transformation).
- Line 263: Clarify Link to Glucose Availability
Response: Following revisions have been included.
“GlcCer synthesis requires UDP-glucose, directly coupling glycolytic activity and glucose availability to glycosphingolipid production and downstream signaling events.”
- Line 288: Expand on Immune Activation and Cancer
Response: Expanded paragraph now links GlcCer accumulation in Gaucher disease to chemokine-driven immune activation, tissue cellularity, oxidative stress, and DNA damage—ultimately creating a pro-tumorigenic microenvironment.
- Line 322: Replace “Passive Lipid”
Response: The Following revisions have been incorporated
“GlcCer functions as an active and versatile lipid mediator, dynamically regulating metabolic and signaling processes relevant to cancer initiation and progression.”
We hope that the revised manuscript is now substantially strengthened in scope and depth and well-suited for publication.
Sincerely,
Manoj Pandey
9-24-2025
Round 2
Reviewer 1 Report
Comments and Suggestions for Authors
Figure Resolution: All your figures are currently too low-resolution. They are quite pixelated on my end and won't pass the journal's quality check. Could you please replace all figures with high-resolution versions (aim for at least 300 DPI)?
Missing Citations for a Key Paragraph: The paragraph discussing the challenges of clinical translation (starting with "Nevertheless, translating these insights...") is making some really important claims about drug delivery, toxicity, and metabolic rewiring. However, there are no references cited throughout that entire section. These points need to be backed up by literature. Please add specific citations to support these statements.
Vague Citation in the TME Section: In your new section on the TME, you've used a block citation [67,96-100] at the very end of the paragraph. This isn't specific enough. That paragraph contains several distinct ideas—cytokine secretion, feedback loops from immune cells, and effects on stromal components. Please go back through that section and place the appropriate reference(s) directly after the specific statement they support. It makes the evidence trail much clearer for the reader.
None
Author Response
We sincerely thank the reviewers for their constructive feedback, which has been invaluable in improving the clarity, rigor, and overall quality of our manuscript. Below, we provide detailed point-by-point responses. We believe that the revisions made fully address the reviewers’ concerns and have substantially strengthened the manuscript for publication.
Reviewer Comment:
Figure Resolution: All your figures are currently too low-resolution. They are quite pixelated on my end and won't pass the journal's quality check. Could you please replace all figures with high-resolution versions (aim for at least 300 DPI)?
Response:
We appreciate the reviewer’s attention to figure quality. In response, we have replaced all figures with high-resolution versions (minimum 500 DPI) to exceed the journal requirements and ensure crisp, publication-ready quality. Each figure has been carefully reviewed to maintain both clarity and data integrity.
Reviewer Comment:
Missing Citations for a Key Paragraph: The paragraph discussing the challenges of clinical translation (starting with "Nevertheless, translating these insights...") is making some really important claims about drug delivery, toxicity, and metabolic rewiring. However, there are no references cited throughout that entire section. These points need to be backed up by literature. Please add specific citations to support these statements.
Response:
We thank the reviewer for weight the need to strengthen this section with supporting literature. We have thoroughly revised the paragraph and incorporated specific, up-to-date citations.
Reviewer Comment:
Vague Citation in the TME Section: In your new section on the TME, you've used a block citation [67,96-100] at the very end of the paragraph. This isn't specific enough. That paragraph contains several distinct ideas—cytokine secretion, feedback loops from immune cells, and effects on stromal components. Please go back through that section and place the appropriate reference(s) directly after the specific statement they support. It makes the evidence trail much clearer for the reader.
Response:
We appreciate this insightful suggestion and fully agree that precise citation placement improves clarity. Accordingly, we have revised the paragraph of TME section by placing individual references immediately after the relevant statements regarding cytokine secretion, immune cell feedback loops, and stromal modulation. These changes create a clear and traceable evidence trail and significantly enhance the scientific rigor of this section.
I believe that the revised manuscript meets the Journal's quality standards for publication.
Sincerely,
Manoj Pandey
9-30-2025